# *Centipeda minima* Extract Inhibits Inflammation and Cell Proliferation by Regulating JAK/STAT Signaling in Macrophages and Keratinocytes

**DOI:** 10.3390/molecules28041723

**Published:** 2023-02-11

**Authors:** Yuanqiang Ma, Byoung Ha Kim, Sang Kyu Yun, Yoon-Seok Roh

**Affiliations:** 1College of Pharmacy and Medical Research Center, Chungbuk National University, Cheongju 28160, Republic of Korea; 2D. Nature Co., Ltd., Seongnam 13174, Republic of Korea

**Keywords:** CMX, psoriasis, JAK/STAT, keratinocytes

## Abstract

Psoriasis, a chronic inflammation-mediated skin disease, affects 2–3% of the global population. It is characterized by keratinocyte hyperproliferation and immune cell infiltration. The JAK/STAT3 and JAK/STAT1 signaling pathways play an important role in the development of psoriasis when triggered by IL-6 and IFN-γ, which are produced by dendritic cells and T-lymphocytes. Thus, blocking JAK/STAT signaling may be a potential strategy for treating psoriasis. Therefore, we examined the effects of CMX, an extract of *Centipeda minima* enriched in Brevilin A, Arnicolide D, Arnicolide C, and Microhelenin C, on macrophages and keratinocytes. We established an in vitro model of psoriasis, based on an inflammation-associated keratinocyte proliferation model, and used macrophages and keratinocytes treated with LPS, IL-6, or IFN-γ to evaluate the effect of CMX. We found that CMX reduced pro-inflammatory cytokine production, by inhibiting lipopolysaccharide (LPS)-induced JAK1/2 and STAT1/3 phosphorylation in macrophages. Moreover, CMX-downregulated chemokine expression and cell proliferation compared with components in HaCaT cells, induced by rh-IL-6 and rh-IFN-γ, respectively. Consistently, we demonstrated that the reduction in chemokine expression and hyperproliferation was mediated by the regulation of IFN-γ-activated JAK/STAT1 and IL-6-activated JAK/STAT3 signaling. In conclusion, CMX inhibited JAK/STAT-mediated inflammatory responses and cell proliferation in macrophages and keratinocytes. Consequently, CMX may have potential uses as a therapeutic agent for treating psoriasis.

## 1. Introduction

Psoriasis, a chronic inflammation-mediated skin disease, affects 2–3% of the population worldwide and 8–11% of the population in Western countries [1,2,3]. Although psoriasis does not directly cause death, it has a significant detrimental effect on patient quality of life and is associated with some comorbidities. General treatments, such as topical treatments, systemic treatments, and biological agents (infliximab and etanercept), are not sufficient to achieve long-term eradication and serious adverse reactions, although they have varying degrees of efficacy. Therefore, it is necessary to synthesize and develop new drugs to help patients who suffer from psoriasis.

Psoriasis is characterized by keratinocyte hyperproliferation and inflammatory cell infiltration in epidermal tissue. Keratinocytes, immune cells, and other resident skin cells contribute to the development of psoriasis. However, its pathogenesis remains complex and is not yet fully understood. Previous studies have shown that environmental factors play a role in recruiting and activating immune cells, such as epidermal plasmacytoid dendritic cells, and CD4-positive and CD8-positive T cells [4,5]. Activated immune cells release several cytokines, such as IL-23, IL-17, IL-20, IL-22, IL-1β, IL-6, and INF-γ, which play a vital role in the pathogenesis of psoriasis [6]. Skin-resident keratinocytes are subsequently stimulated by these cytokines, exacerbating the effects of cytokine and chemokine activation and proliferation [7,8]. Targeting the secretion of cytokines and chemokines in immune cells has helped to make great strides in the treatment of psoriasis. Notably, the complexity of epidermal tissues and the limitations of targeted drugs have motivated researchers to explore novel therapeutic strategies.

The JAK/STAT1 signaling pathway is involved in a range of autoimmune diseases, including rheumatoid arthritis [9], inflammatory bowel disease [10], hepatitis [11], and psoriasis [12]. Various cytokines stimulate different isoforms of JAK protein kinases, namely JAK1, JAK2, and JAK3, which then activate STAT transcription factors to target a series of genes. STAT1 and STAT3 have recently emerged as key players in the development and pathogenesis of psoriasis [13,14]. In immune cells activated by chronic inflammation, the transcription of STAT1 and STAT3 promotes cytokine secretion during psoriasis. The simultaneous activation of dendritic cells and differentiation of T-lymphocytes intensifies the immune response. In this complex immune network, IL-6, IL-23, and Interferon gamma (IFN-γ) are directly involved in the proliferation and differentiation of keratinocytes [15]. In addition, JAK/STAT dysfunction has been associated with polycythemia vera, essential thrombocytopenia, and severe combined immunodeficiency and hyperimmunoglobulin E syndrome. Furthermore, keratinocytes are affected by IL-22, IL-19, IL-20, and IL-24. In fact, JAK1 and JAK2 are regulated by these factors to promote the activation of downstream STAT-transcription factors. Thus, the inhibition of JAK/STAT signaling has become a potential therapeutic option for treating psoriasis.

CMX (*Centipeda minima* extract) is enriched with Brevilin A, Arnicolide C, Arnicolide D, and Microhelenin C. Brevilin A is a sesquiterpene lactone from Centipeda minima and exhibits various biological activities. Brevilin A has a strong inhibition on STAT1 and STAT3 signaling pathway-dependent target gene expression and cell proliferation [16]. Arnicolide is another type of sesquiterpene lactone, including Arnicolide C and Arnicolide D. Arnicolide D can downregulate cell cycle proteins and promote cell apoptosis through PI3K/AKT/mTOR and STAT3 signaling pathways [17]. Microhelenin C, as one of the ingredients in CMX, has a synergistic function in promoting hair regrowth and hair loss treatment [18]. Their chemical structures are shown in Figure 1. It has been reported that these compounds have anti-inflammatory, anticancer, anti-allergy, and antiproliferative properties when used alone. In this study, we firstly clarified the synergistic effect of these four compounds, that is, Brevilin A, Arnicolide C, Arnicolide D and Microhelenin C, in an in vitro model of psoriasis. Our findings showed, that the CMX mixture can inhibit the inflammatory response in macrophages and the proliferation of keratinocytes. Most importantly, the CMX mixture had a stronger inhibitory effect on the JAK/STAT pathway in the mechanism.

## 2. Results

### 2.1. CMX Analysis

Four chemical markers in five batches of CMX samples were quantified using HPLC-UV. The representative chromatogram of CMX is shown in Figure 2A. The retention time of Arnicolide D, Arnicolide C, Microhelenin C, and Brevilin A were 12.5, 14.3, 17.2, and 19.3 min, respectively, and the content of each compound in the CMX fraction was 14.92 ± 4.25, 12.24 ± 7.58, 2.75 ± 0.76, and 62.93 ± 17.00 mg/mL, respectively. The content of Brevilin A in CMX was two times higher than that of the other three combined, and Brevilin A covered 67.78% of the total area (Figure 1). The chromatographic results and contents are summarized in Table 1.

### 2.2. CMX Inhibits Pro-Inflammatory Factors Production in Macrophages Induced by LPS 

Studies have found that toll-like receptor (TLR) 4 plays a key role in psoriasis [19]. In order to investigate whether CMX inhibits the macrophage inflammatory response in vitro, we used LPS to induce inflammation in the mouse macrophage cell line RAW264.7. In LPS-stimulated RAW 264.7 cells, there was a significant increase in the expression of the inflammatory cytokines and chemokines IL-6, IL-1β, TNF-α, CXCL1, CCL2, CCL3, IFN-β, CXCL10, and CCL20 mRNA. Notably, CMX inhibited this expression (Figure 3). CMX inhibited IL-6 and TNF-α mRNA expression more effectively than Arnicolide D alone; it also inhibited CCL20 mRNA expression more effectively than Arnicolide C alone. Furthermore, CMX had a stronger inhibitory effect on IL-6, TNF-α, and CCL20 than Arnicolide D and Arnicolide C when used alone. It seems that the CMX mixture did not significantly suppress the inflammatory response in macrophages induced by LPS, compared with the use of these components alone, so we will further examine the protein levels in the JAK/STAT pathway.

### 2.3. CMX Blocks LPS-Induced JAK/STAT Signaling Pathway in Macrophages

We found that the CMX mixture and its components downregulated the expression of pro-inflammatory genes. To explore how CMX inhibited the expression of mRNAs for pro-inflammatory cytokines and chemokines in macrophages, we hypothesized that CMX inhibits the JAK/STAT signaling pathway to hinder the transcription of pro-inflammatory factors. To test this hypothesis, we used LPS to induce the phosphorylation of JAK1, JAK2, and STAT3, as well as the expression of ERK protein in RAW264.7 cells. We found that CMX had a stronger capacity to inhibit JAK1 and JAK2 phosphorylation than either Arnicolide C or Microhelenin C alone (Figure 4A). Therefore, CMX had the strongest inhibitory effect on phosphorylated STAT3 compared with the other components alone. Additionally, CMX downregulated the phosphorylation levels of STAT1 and ERK compared with the other components alone. (Figure 4B). 

### 2.4. CMX Suppresses Inflammation-Induced Keratinocytes Proliferation through JAK1/2-STAT3

We have found that CMX and its components significantly reduced the pro-inflammatory response through the JAK/STAT signaling pathway. Therefore, we should focus on another pathogenic factor of psoriasis. Psoriasis is a kind of skin disease mediated by chronic inflammation accompanied by the hyperproliferation of keratinocytes. We found that IL-6 is a cytokine, critical for skin wound healing, and a pathogenic factor for the abnormal proliferation of keratinocytes [8,20]. To further explore the effect of CMX on the pathogenesis of psoriasis, we cultured HaCaT cells and incubated them with IL-6 to mimic the abnormal proliferation of keratinocytes in vitro. We observed that IL-6 significantly induced the hyperproliferation of keratinocytes at 12, 24, 48, and 72 h (Figure 5A). We also found that CMX had a significant inhibitory effect on cell proliferation compared with these components after 24 h treatment. At the mechanistic level, we examined the JAK/STAT signaling pathway. We determined that IL-6-induced keratinocyte proliferation occurred through the JAK/STAT3 signaling pathway. Additionally, it was determined that CMX significantly inhibited the IL-6-induced phosphorylation of JAK1, JAK2, and STAT3 (Figure 5B).

### 2.5. CMX Suppresses Inflammation-Induced Chemokines Production in Keratinocytes via JAK1-STAT1

We found that CMX reduced the proliferation of keratinocytes induced by IL-6. At the same time, keratinocytes hyperproliferation promoted the secretion of chemokines, and thereby recruited immune cells to devote into the development of psoriasis. Therefore, we explored the effects of CMX on chemokine production in keratinocytes. Chemokines produced by keratinocytes are crucial factors in the recruitment and activation of T-lymphocytes during psoriasis progression [21]. Other immune cells also play important roles in the secretion of keratinocyte chemokines. IFN-γ is involved in the pathogenesis of psoriasis by activating JAK1 and STAT1 transcription factors in skin lesions [22]. IFN-γ plays an important role in inducing the inflammatory response of keratinocytes. Therefore, we treated the keratinocytes with IFN-γ. The results showed that IFN-γ significantly induced mRNA expression of keratinocytes chemokines CCL17, CXCL9, CXCL10, and IFIT2. Additionally, CMX significantly suppressed the expression of mRNA for chemokines CCL17, CXCL9, CXCL10, and IFIT2 compared with other components (Figure 6A). Furthermore, we determined the JAK/STAT signaling pathway. We found that IFN-γ induced chemokines mRNA expression by JAK1/STAT1. Specifically, the results showed that CMX significantly inhibited the IFN-γ-mediated activation of the JAK1/STAT1 pathway (Figure 6B).

## 3. Discussion

Psoriasis is a chronic autoimmune disease characterized by the formation of scaly, indurated, and erythematous plaques [23]. With the infiltration of immune cells, the inflammatory response gradually intensifies, and skin keratinocytes are seriously challenged by the production of cytokines. Notably, in this context, the intracellular non-receptor tyrosine kinase JAK acts as an important signaling bridge that mediates pathogenic factors. These pathogenic factors include IL-6, IL-17, IL-23, and IFN-γ [15]. There is also growing evidence that JAK proteins are potential therapeutic targets for psoriasis and other immune-mediated skin diseases [9,24].

In this study, we explored the effects of CMX on psoriasis pathogenesis in vitro. We found that CMX significantly downregulated the expression of pro-inflammatory genes. Furthermore, we observed that CMX significantly inhibited the expression of IL-6, IL-1β, TNF-α, CCL3, IFN-β, CXCL10, and CCL20 mRNA when compared to any other single component tested. Although the CMX mixture did not significantly inhibit cytokines, mRNA expression compared to Brevilin A and Microhelenin C alone, the protein level showed that the CMX mixture reduced the expression of p-JAK1/JAK2 more significantly than Microhelenin C alone. This may be attributed to the high level of Brevilin A and the synergistic effect of each component. The targeted activation of macrophages, dendritic cells, and T-lymphocytes may play a crucial role in psoriasis pathogenesis. Therefore, we firstly focused on the role of the CMX mixture in pro-inflammatory inflammation. Surprisingly, both CMX and its components reduced the expression of pro-inflammatory genes. However, there were no significant differences when compared with each component alone. We speculate that under the absence of toxic effects, other physiological functions may be affected without significantly reducing the inflammatory response.

Recent studies have shown that epidermal hyperproliferation, differentiation, and inflammatory responses depend on STAT3 activation in the IMQ-mediated psoriasis model [25,26]. Therefore, STAT3 may be a potential target for psoriasis therapy. To investigate this possibility, we activated STAT3 in vitro to mimic the chronic inflammation model of psoriasis. Our study found that CMX blocked the activity of the JAK1/JAK2-STAT3 axis. Notably, we found that the CMX mixture inhibited STAT1 or STAT3 phosphorylation more significantly than individual components. Shuna et al. confirmed that 2′-hydroxycinnamaldehyde, isolated from the stem bark of *Cinnamomum cassia*, ameliorated T-cell activation and keratinocyte hyperproliferation in vivo by inhibiting the activation of STAT3 [27]. Furthermore, Lihua et al. found that berberine interfered with the JAK-STAT3 signaling pathway, and ameliorated pathological changes in vivo [25]. This finding is similar to ours. Therefore, we concluded that the CMX alleviates the production of pro-inflammatory factors by blocking the JAK1/2-STAT3 signaling pathway.

Based on the effect of CMX on the JAK/STAT pathway in macrophages, we investigated its role in keratinocytes. Previous studies have shown that IL-17A, IL-22, and IL-6 drive the hyperproliferation of keratinocytes [20,28]. We found that the CMX mixture significantly downregulated cell proliferation within 12, 24, 48 and 72 h at the level of IL-6-mediated keratinocytes proliferation. This compensated for the fact that CMX did not significantly decrease the mRNA expression of inflammatory cytokines relative to Brevilin A and Microhelenin C. This is most likely due to differences in the concentration of CMX and other components. The downregulation of the CMX-based inflammatory response was involved in the JAK/STAT signaling pathway. Additionally, in the pathogenesis of psoriasis, JAK/STAT protein is abnormally expressed. We speculated that CMX also affects the physiological functions of skin-resident cells called keratinocytes. Furthermore, the mechanism for the inhibition of cell proliferation involves the inactive JAK1/2-STAT3 pathway. Most importantly, CMX showed a stronger inhibitory activity to keratinocytes proliferation, induced by IL-6, than any of the other ingredients alone. This result indicated that the synergistic effect enhanced the inhibitory activity of different components. We found that IL-6 promoted keratinocyte proliferation over time. CMX significantly downregulated cell proliferation after 24, 48, and 72 h. This seems to be related to the promotion of cell proliferation. Excessive cell proliferation will promote keratinocytes dysfunction and, in particular, the secretion of chemokines, which is the key to maintaining long-term chronic inflammation in psoriasis. Therefore, we tried to use IFN-γ to induce the inflammatory response in keratinocytes. We found that CMX downregulated IFN-γ-mediated keratinocyte chemokine mRNA production. Of course, this is inseparable from the activation of the JAK1/STAT1 signaling pathway. Our results showed that CMX more strongly downregulated JAK2 phosphorylation protein levels compared to JAK1. This appears to have a similar effect to limiting keratinocytes proliferation. Interestingly, although the CMX mixture significantly inhibited cell proliferation, it did not downregulate inflammatory factors compared to Brevilin A and Microhelenin C. This suggests that the CMX mixture has a cooperative function between immune response and cell differentiation. This synergistic effect may be very applicable in psoriasis with complex pathogenesis. Notably, we believe that the CMX mixture has a significant inhibitory effect on the JAK/STAT axis. Based on the above results, we found that the CMX mixture has a better collaborative effect on inhibition between inflammation and cell proliferation. Next, we plan on determining the pharmacological activity of CMX in a mouse model of psoriasis.

## 4. Materials and Methods

### 4.1. Cells Culture

The RAW264.7 cells, a macrophage-like cell line derived from BALB/c mice and HaCaT cells, spontaneously transformed aneuploid immortal keratinocyte cell line from adult human skin cultured in Dulbecco’s Modified Eagle’s Medium (DMEM) (Corning^®^, Glendale, CA, USA) supplemented with 10% fetal bovine serum (FBS) (Corning^®^, Glendale, CA, USA) and 1% penicillin/streptomycin (Corning^®^, Glendale, CA, USA) or 1% antibiotic-antimycotic solution (Gibco™, Waltham, MA, USA) in a 5% CO_2_ atmosphere at 37 °C. In order to evaluate the relevant gene mRNA levels and the anti-inflammatory effects of CMX. RAW264.7 and HaCaT cells (2.5 × 10^5^ cells/mL) were treated with lipopolysaccharide (LPS, 125 ng/mL) (Sigma-Aldrich, St. Louis, MO, USA), rh-IFN-γ (10 ng/mL), CMX (3 µg/mL), Brevilin A, Arnicolide D, Arnicolide C, and Microhelenin C for 6 h at 37 °C. In order to measure the relevant protein expression, RAW264.7 and HaCaT cells (4.5 × 10^5^ cells/mL) were pre-incubated with CMX for 1 h before being co-cultured with LPS and rh-IL-6 (10 ng/mL), or rh-IFN-γ (10 ng/mL) for 1 h at 37 °C. CCK-8 was used to measure cell proliferation assay in HaCaT cells (8 × 10^4^ cells/mL). HaCaT cells were incubated by CMX (6 µg/mL) for 0, 12, 24, 48, and 72 h.

### 4.2. Quantitative Real-Time Polymerase Chain Reaction (RT-qPCR) Analysis

Cells were homogenized in RiBoEx (Geneall Biotechnology Co., Ltd., Seoul, Korea). Cell lysates were mixed with 200 ul chloroform and centrifuged at 12,000× *g* for 15 min at 4 °C. Total RNA was determined by performing phase separation, binding, washing, and elution using the Hybrid-R™ kit (Geneall Biotechnology Co. Ltd. Seoul, Republic of Korea) as per the manufacturer’s instructions. Subsequently, primer Script TM RT Reagent Kit with Genomic DNA (gDNA) Eraser (TAKARA Bio Inc. Shiga, Japan) was used for complementary DNA (cDNA) reverse transcription. RT-qPCR was performed using the CFX Connect Real-Time PCR Detection System (Bio-Rad, Hercules, CA, USA). The cycling conditions were as follows: enzyme activation for 1 min at 95 °C, denaturation for 15 s at 95 °C, and annealing for 15 s at 60 °C. qRT-PCR was performed for at least 45 cycles. The targeted mRNA expression genes were normalized to glyceraldehyde 3-phosphate dehydrogenase (Actin, internal control) expression and were calculated using the 2^−ΔΔCt^ method. The specific primer sequences are listed in Table 2. 

### 4.3. Western Blot Analysis

Cells were washed twice with phosphate-buffered saline (PBS), collected from plates, and lysed on ice using radioimmunoprecipitation assay buffer (RIPA) plus a phosphatase inhibitor (Halt™ Protease and Phosphatase Inhibitor Cocktail, EDTA-free (100×), Thermo Fisher Scientific, Waltham, MA, USA) for 30 min, and vortexed for 10 min. Cell lysates were performed by centrifugation at 13,000 RPM at 4 °C for 15 min. Supernatant was collected and the protein concentration was detected using the BCA Protein Assay Kit (Thermo Fisher Scientific Inc., Waltham, CA, USA). All protein samples were prepared in 5X loading buffer and heated at 95 °C for 10 min. Subsequently, an equal amount of protein was separated by 10% SDS-PAGE and transferred to PVDM (polyvinylidene difluoride membranes). Membranes were blocked with 5% bovine serum albumin (BSA) solution at 4 °C overnight. Membranes were incubated by first antibodies (ABclonal, MA, USA; Santa Cruz, CA, USA; or Sigma-Aldrich, St. Louis, MI, USA) (1: 1000). p-JAK1 (#AP0530), JAK1 (#A0715), p-JAK2 (#AP0531), JAK2 (#A11497), p-STAT1 (#AP1000), STAT1 (#A19563), p-STAT3 (#AP0705), STAT3 (#A1192), p-ERK (#SC-7383), ERK (#SC-514302), and β-actin (#A5441) antibodies were diluted to 1:1000, at 4 °C overnight. After washing with Tris-buffered saline with 0.1% Tween^®^ 20 Detergent (TBST, PH 7.4) 3 times for 10 min, horseradish peroxidase-conjugated secondary antibodies (Thermo Fisher Scientific, MA, USA) were kept at room temperature for 1 h. Protein signaling was detected with West-Q Pico Dura ECL Solution (Thermo Fisher Scientific, MA, USA), according to manufacturer’s instructions. Finally, protein signaling was quantified by ImageJ (Version 1.52, NIH, Bethesda, MD, USA) and normalized to β-actin. All antibodies were diluted in tris-buffered saline/Tween containing 5% BSA. 

### 4.4. Cell Counting Kit-8 (CCK-8) Proliferation Assay

HacaT cells were cultured at a density of 1500 cells per well, in a 96-well plate overnight. CMX, Brevilin A, Arnicolide D, Arnicolide C, and Microhelenin C were pre-treated for 1 h, and then co-incubated with rh-IL-6 for 12 h, 24 h, 48 h, and 72 h in a CO_2_ incubator. Cell proliferation was determined by CCK-8 kit (Dojindo, Kumamoto, Japan). In total, 10 µL CCK-8 reagent was added into every well, and the cell culture plate was incubated at 37 °C for 30 min. The optical density (OD) values were determined at 450 nm using a microplate reader. Proliferation was performed in quadruplicate.

### 4.5. Plant Materials and Preparation of CMX

Centipeda minima was purchased in December 2019 from Natural herb (Goesan, Korea). Centipeda minima extract (CMX^®^) was prepared by D. Nature Co., Ltd. (Seongnam, Republic of Korea) by utilizing the efficient separation of Brevilin A, Arnicolide D, Arinicolide C, and Microhelenin C from Centipeda minima by induc-ing phase separation in the emulsion.

### 4.6. Statistical Analysis

Data are expressed as the means ± SEM from at least three independent experiments. GraphPad Prism 7 software (GraphPad Software Inc., San Diego, CA, USA) was used to perform statistical analysis with the unpaired Student’s t test. * *p* < 0.05, ** *p* < 0.01, and *** *p* < 0.001 were considered statistically significant.

## 5. Conclusions

In conclusion, CMX exhibited anti-inflammatory and antiproliferative effects in an in vitro model of psoriasis. CMX enriched with Brevilin A, Arnicolide D, Arnicolide C, and Microhelenin C had the strongest inhibitory effect against cytokines and chemokines and inhibited the phosphorylation level of the JAK/STAT protein. Therefore, CMX may be a potentially therapeutic agent for the treatment of psoriasis.

## Figures and Tables

**Figure 1 molecules-28-01723-f001:**
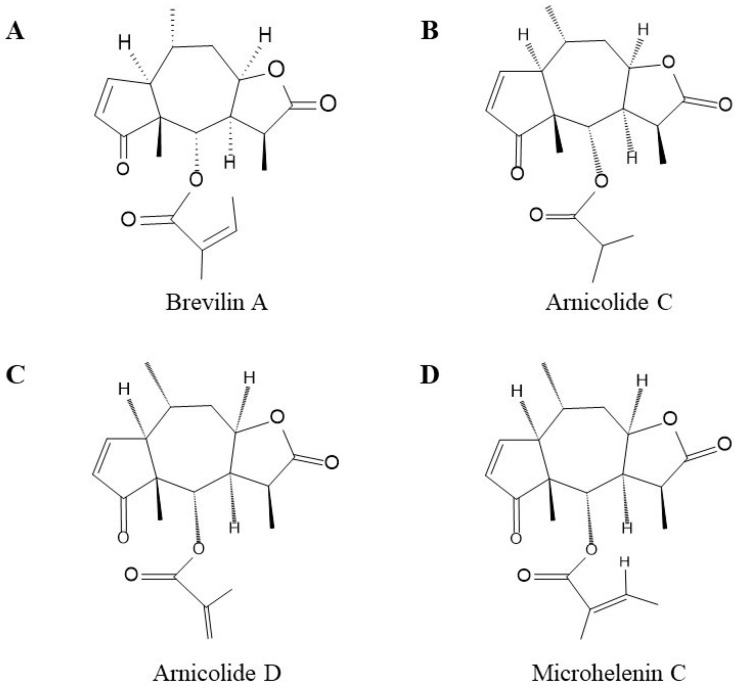
Chemical structure of CMX components. (**A**) Brevilin A. (**B**) Arnicolide C. (**C**) Arnicolide D. (**D**) Microhelenin C.

**Figure 2 molecules-28-01723-f002:**
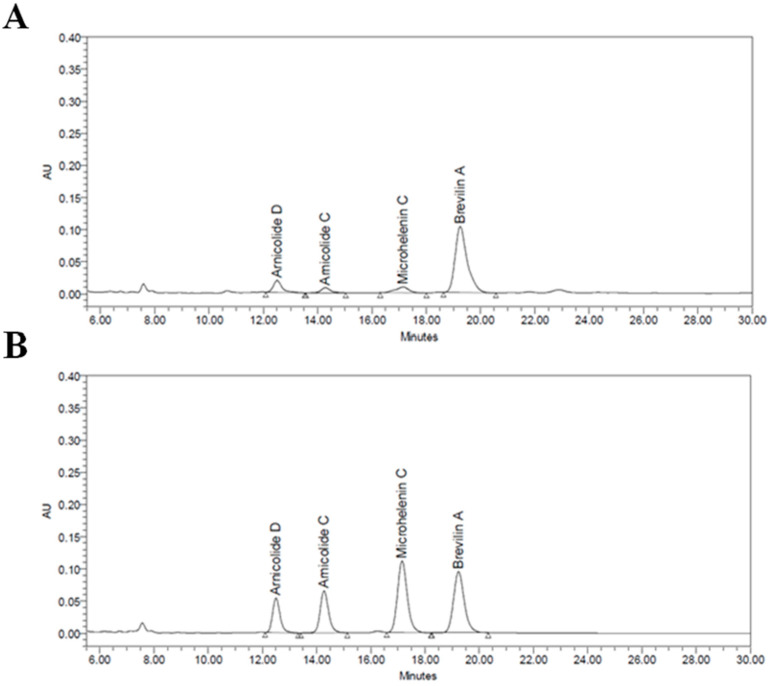
HPLC chromatogram of CMX. (**A**) HPLC chromatogram of CMX sample. (**B**) HPLC chromatogram of standard mixtures of Aricolide D, Arnicolide C, Microhelenin C, and Brevilin A. CMX, emulsion extract from Centipeda minima; HPLC, high-performance liquid chromatography. The column temperature was maintained at 40 °C and the wavelength was set at 224 nm.

**Figure 3 molecules-28-01723-f003:**
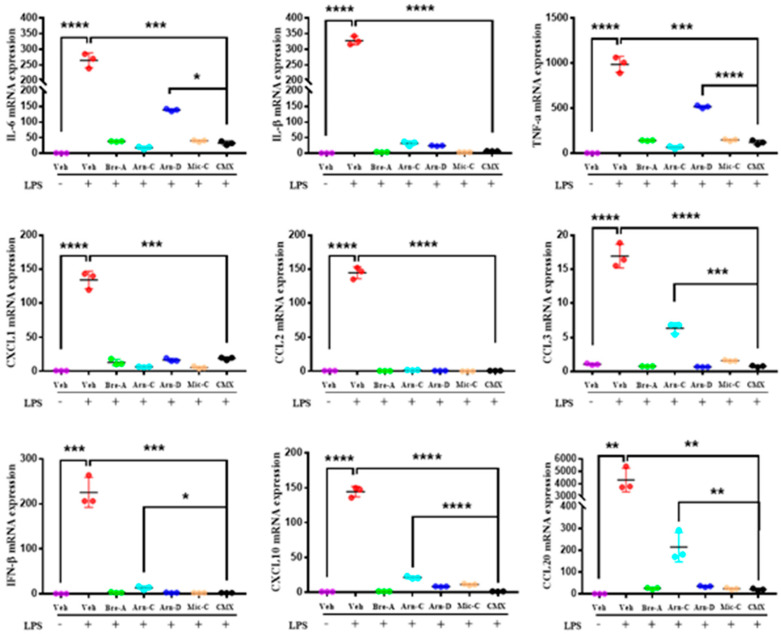
CMX inhibits pro-inflammatory factors production in macrophages induced by LPS. The mRNA expression of IL-6, IL-1β, TNF-α, CXCL1, CCL2, CCL3, IFN-β, CXCL10, and CCL20 were detected in macrophages by RT-qPCR. Relative mRNA expression levels were normalized to mouse GAPDH levels. The data were expressed as means ± sem. * *p* < 0.05, ** *p* < 0.01, *** *p* < 0.001, **** *p* < 0.0001. Three repeated trials were presented per group (*n* = 3). All concentrations of mixtures or components are 3 µg/mL. Bre-A, Brevilin A; Arn-C, Arnicolide C; Arn-D, Arnicolide D; Mic-C, Microhelenin C; CMX, emulsion extract from Centipeda minima.

**Figure 4 molecules-28-01723-f004:**
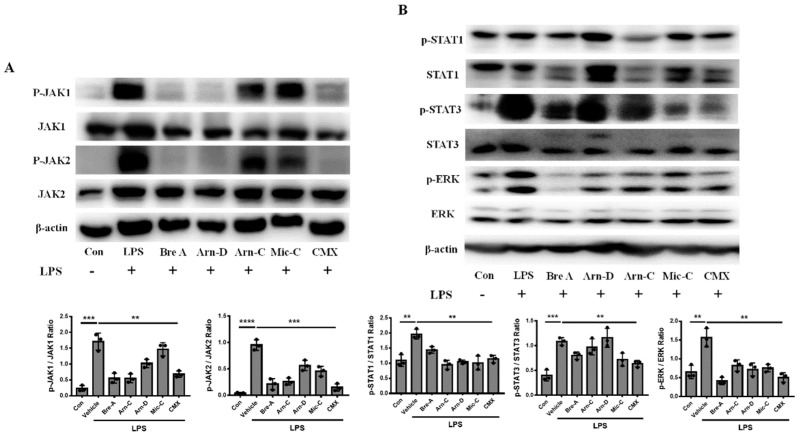
CMX blocks LPS-induced JAK/STAT signaling pathway in macrophages. (**A**) The protein expression of p-JAK1, JAK1, p-JAK2, JAK2, and β-actin was evaluated by Western blot. (**B**) The protein expression levels of p-STAT1, STAT1, p-STAT3, STAT3, p-ERK, ERK, and β-actin were detected after treatment with or without LPS, and co-treatment with Brevilin A, Arnicolide D, Arnicolide C, Microhelenin C, or CMX with LPS. All the concentrations of mixtures or components were 3 µg/mL. The data were expressed as means ± sem. ** *p* < 0.01, *** *p* < 0.001, **** *p* < 0.0001. All experiments were repeated 3 times. Bre-A, Brevilin A; Arn-D, Arnicolide D; Arn-C, Arnicolide C; Mic-C, Microhelenin C; CMX, emulsion extract from Centipeda minima.

**Figure 5 molecules-28-01723-f005:**
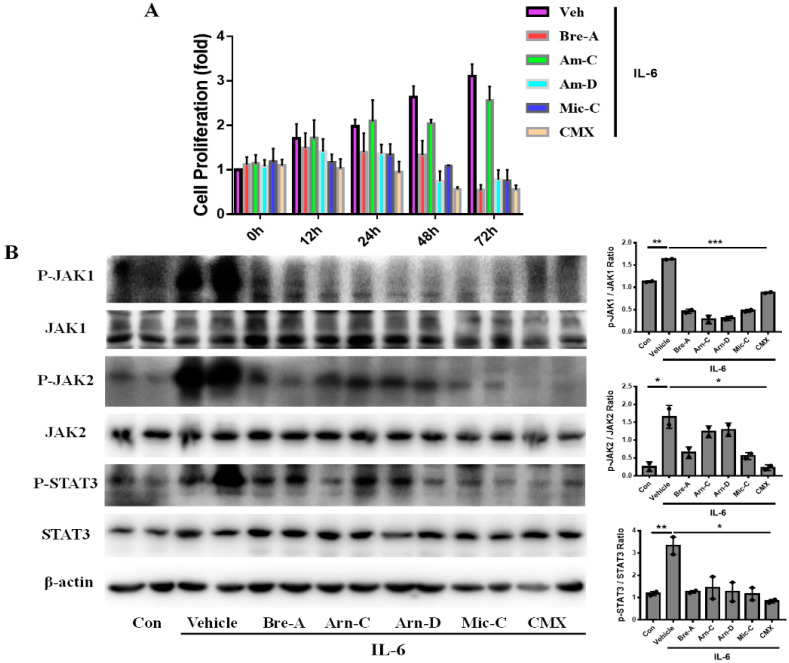
CMX suppresses keratinocytes proliferation through JAK1/2-STAT3 pathway induced by IL-6. (**A**) Cell proliferation was detected in HaCaT incubated by only rh-IL-6 or co-incubated with CMX for 0, 12, 24, 48, and 72 h. (**B**) Protein expression of p-JAK1, JAK1, p-JAK2, JAK2, p-STAT3, and STAT3 was evaluated by Western blot. The data were expressed as means ± sem. * *p* < 0.05, ** *p* < 0.01, *** *p* < 0.001. Two repeated trials were presented per group (*n* = 2). All experiments were repeated 3 times. All the concentrations of mixtures or components were 3 µg/mL or 6 µg/mL. Bre-A, Brevilin A; Arn-D, Arnicolide D; Arn-C, Arnicolide C; Mic-C, Microhelenin C; CMX, emulsion extract from Centipeda minima.

**Figure 6 molecules-28-01723-f006:**
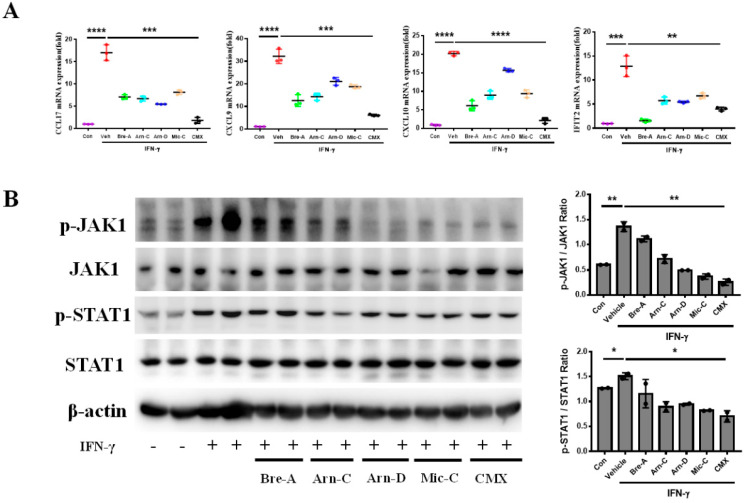
CMX suppresses keratinocytes chemokines production via JAK1-STAT1 pathway induced by IFN-γ. (**A**) The mRNA expression of CCL17, CXCL9, CXCL10, and IFIT2 was detected in HaCaT by RT-qPCR. Relative mRNA expression levels were normalized to mouse GAPDH levels. The data were expressed as means ± sem. * *p* < 0.05, ** *p* < 0.01, *** *p* < 0.001, **** *p* < 0.0001. Three repeated trials were presented per group (*n* ≥ 3). (**B**) Protein expression of p-JAK1, JAK1, p-STAT1, and STAT1 was evaluated by Western blot. Two repeated trials were presented per group (*n* = 2). All experiments were repeated 3 times. All concentrations of mixtures or components were 6 µg/mL or 3 µg/mL. Bre-A, Brevilin A; Arn-D, Arnicolide D; Arn-C, Arnicolide C; Mic-C, Microhelenin C; CMX, emulsion extract from Centipeda minima.

**Table 1 molecules-28-01723-t001:** Contents of Arnicolide D, Arnicolide C, Microhelenin C, and Brevilin A in CMX.

Compounds	Arnicolide D	Arnicolide C	Microhelenin C	Brevilin A
RT (min)	12.5	14.3	17.2	19.3
Content (mg/mL)	14.92 ± 4.25	12.24 ± 7.58	2.75 ± 0.76	62.93 ± 17.00

CMX, extract from Centipeda minima; RT, retention time.

**Table 2 molecules-28-01723-t002:** Primer sequences of the genes used for quantitative real-time polymerase chain reaction.

Gene	Forward (5′ to 3′)	Reverse (5′ to 3′)
IL-6	ACAAAGCCAGAGTCCTTCAGAGAG	TTGGATGGTCTTGGTCCTTAGCC
IL-1β	CTCGCAGCAGCACATCAACAAG	CCACGGGAAAGACACAGGTAGC
TNF-α	AGGGTCTGGGCCATAGAACT	CCACCACGCTCTTCTGTCTA
CXCL1	TGCACCCAAACCGAAGTC	GTCAGAAGCCAGCGTTCACC
CCL2	ATTGGGATCATCTTGCTGGT	CCTGCTGTTCACAGTTGCC
CCL3	TCCCAGCCAGGTGTCATTTTCC	CAGTTCCAGGTCAGTGATGTATTCTTG
CXCL10	GACGGTCCGCTGCAACTG	CTTCCCTATGGCCCTCATTCT
CCL20	GATCCAAAGCAGAACTGGGTGAA	GGACAAGTCCACTGGGACACAA
IFN-β	ACAGCCCTCTCCATCAACTATAAGC	GCATCTTCTCCGTCATCTCCATAGG
CCL17	TGAGGACTGCTCCAGGGATG	AACGGTGGAGGTCCCAGGTA
CXCL9	GCAGCCAAGTCGGTTAGTGGA	TTAAATTCTGGCCACAGACAACCTC
CXCL10	GGCCATCAAGAATTTACTGAAAGCA	TCTGTGTGGTCCTTGGAA
IFIT2	CACAGGTGTGAACCAAATCCAAATA	GTCAATGGTAGCAGGTGGCAGA
Actin	CATCCGTAAAGACCTCTATGCCAAC	ATGGAGCCACCGATCCACA

IL-6, Interleukin 6; IL-1β, Interleukin-1 beta; TNF- α, Tumor necrosis factor alpha; CXCL1, Chemokine (C-X-C motif) ligand 1; CXCL10, C-X-C motif chemokine ligand 10; CXCL9, Chemokine (C-X-C motif) ligand 9; CCL2, Chemokine (C-C motif) ligand 2; CCL3, Chemokine (C-C motif) ligand 3; CCL20, Chemokine (C-C motif) ligand 20; CCL17, CC Chemokine ligand 17; IFN-β, Type-I interferon beta; IFIT2, Interferon-induced protein with tetratricopeptide repeats 2.

## Data Availability

Not applicable.

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
