# Peer review of "Centipeda minima Extract Inhibits Inflammation and Cell Proliferation by Regulating JAK/STAT Signaling in Macrophages and Keratinocytes"

_molecules, 2023, doi:10.3390/molecules28041723_

Round 1

Reviewer 1 Report

This manuscript seems interesting. However, following points should be considered.

1.        In figure legends, the concentrations of mixtures or components should be clearly indicated.

2.        No cell toxicity of mixtures or components were showed in those cells under regular culturing condition. The mixture or components at tested concentrations may directly affect cell proliferation without inflammation stimulation.

3.        In figure legends, sample number should be indicated. In addition, sample number should not be less than 3.

4.        Figure 4A and 4B, Figure 5B and 6B, protein levels have not been quantified and conducted a statistical analysis.

5.        Different inflammatory factor mRNAs were assayed in different cell types. Please indicated why the authors selected these different inflammatory factors.

6.        Please provide vendors of the purchased antibodies or other reagents.

7.        In the discussion section, components in mixtures did not be discussed about their activities.

8.        The authors claimed that “In conclusion, CMX exhibited anti-inflammatory and anti-proliferative effects in vitro a model of psoriasis.”. Please provide a proof or reference to confirm this model is a model of psoriasis.

Author Response

Dear reviewer: 

Thank you very much for your review of our study.

We have completed the revision about manuscript and point-by-point response. Please see the attachment.

Best Wishes

Reviewer 2 Report

The manuscript entitled " Centipeda minima extract inhibits inflammation and cell proliferation by regulating JAK/STAT signaling in macrophages and 3 keratinocytes " is an interesting attempt to gain more information about psoriasis research.

 The overall composition of the manuscript is good, with a very nice introduction to the topic. The quality of the figures and tables is also good. The paper is written in a clear and understanding manner and the results are properly presented. The authors examined the effects of CMX, an extract of Centipeda minima on macrophages and keratinocytes to see if blocking JAK/STAT signaling may be a strategy for treating psoriasis. Their research have shown that CMX may have potential as a therapeutic agent for treating psoriasis because it reduced pro-inflammatory cytokine production by inhibiting Lipopolysaccharide 20 (LPS)-induced JAK1/2 and STAT1/3 phosphorylation in macrophages and downregulated chemokine expression and cell proliferation in HaCaT cells.

Overall I have find this manuscript suitable for publication in “Molecules” in current form.

Author Response

Dear reviewer: 

Thank you very much for your review of our study. 

Best Wishes

Reviewer 3 Report

The research article by Ma Yuan Qiang  et al had checked whether synergetic effect of Centipeda minima extract (CMX) as crude extract would be more potent as compared to each individual content ( Brevilin A, Arnicolide D, 16 Arnicolide C, and Microhelenin C), to inhibits the inflammation and cell proliferation in mouse macrophages and human keratinocytes cells by regulating JAK/STAT signaling.

The researchers have chosen most appropriate model to answer the question. However, the data which is presented contradicts most of the conclusion drawn by the authors

The first question is the purity of the extract. The HPLC chromatogram (Fig 1A) does not show clear peaks for Arnicolide D Arnicolide C Microhelenin C and except Brevilin A as compared to standards which very clear peaks. Moreover, the authors did not show the source of these standards and also, they did not mention from where they got the pure Arnicolide D Arnicolide C Microhelenin C and Brevilin A. Unless the purity of the compounds is not satisfied, the all data presented here is questionable.

2- The raw blot images are the same images as presented in the MS. The raw blot should be full image of membrane including the protein marker to judge the correct protein size of the tested protein. The authors should also mention the source of the antibodies along with cat#, The authors mentioned only the cat#.

3- The authors claimed that there was significant decrease in the expression of the inflammatory cytokines and chemokines IL-6, IL-1β, TNF-α, CXCL1, CCL2, CCL3,  IFN-β, CXCL10, and CCL20 mRNA by CMX  (Fig 3A) as compared to individual contents of the extract. The quality of the images for real time RT PCR graphs are not enough to judge this claim however, the WB clearly shows that he inhibitory of phosphorylated JAK1 activity of Bre A and ARn is much stronger then CMX, whereas CMX effect is almost similar to inhibit of JAK2 phosphorylation. Similarly, Arn C shows more inhibitory activity of p-STAT1 then CMX whereas the expression of  P-STAT3 is same by ArnC and CMX exposure of mouse macrophage cells.

4- the authors have shown that IL-6 induced keratinocytes had higher proliferation after 48 and at 72 hours of induction (Fig 5A) whereas, CMX suppresses inflammation-induced keratinocytes proliferation, In this context the inhibitory effect of CMX was almost similar with each individual contents and there is no significant difference between CMX as crude extract and each individual content. The Fig 5B does not show the unphosphorylated STAT3.

5- The authors detected the effects of CMX on chemokine production in keratinocytes via JAK1- STAT1 after induction of human keratinocytes cell lines with IFN-γ-. As the mRNA expression data are of very low resolution, but the WB (Fig 6 B) showed that there is no significant difference between CMX as crude extract and all other individual contests.

The who manuscript need major revision and authors should present the data which should be satisfactory to accept their claim

Author Response

(The authors gave the same response as above.)

Round 2

Reviewer 1 Report

This manuscript has been carefully revised and can be acceptable to publish in this journal.

Author Response

Dear reviewer:

We greatly appreciate your suggestions and questions about our manuscript.
let me know if there are any questions

Best wishes

Thank you so much

Roh

Reviewer 3 Report

The comments provided by the authors are unsatisfactory. The authors must response to the specific points which are raised. For the correct measurement of band intensity in WB, the authors can use software like image J.

Author Response

Dear reviewer:

We greatly appreciate your suggestions and questions about our manuscript. Based on your comments, we performed a PCR check again and statistical analysis using Image J software.
Looking forward to further communication with you about the manuscript
Please let me know if there are any questions.

Thank you so much

Best wishes

Roh

Round 3

Reviewer 3 Report

The manuscript has been improved significantly, still there are some areas which could have been improved, however i recommend to accept the manuscript for publication in its present form